# Experimental heatwaves compromise sperm function and cause transgenerational damage in a model insect

Kris Sales[1], Ramakrishnan Vasudeva [1], Matthew E. Dickinson [1], Joanne L. Godwin[1], Alyson J. Lumley[1], Łukasz Michalczyk [2], Laura Hebberecht[1], Paul Thomas[1], Aldina Franco [3] & Matthew J.G. Gage[1]

Climate change is affecting biodiversity, but proximate drivers remain poorly understood. Here, we examine how experimental heatwaves impact on reproduction in an insect system. Male sensitivity to heat is recognised in endotherms, but ectotherms have received limited attention, despite comprising most of biodiversity and being more influenced by temperature variation. Using a flour beetle model system, we find that heatwave conditions (5 to 7 °C above optimum for 5 days) damaged male, but not female, reproduction. Heatwaves reduce male fertility and sperm competitiveness, and successive heatwaves almost sterilise males. Heatwaves reduce sperm production, viability, and migration through the female. Inseminated sperm in female storage are also damaged by heatwaves. Finally, we discover transgenerational impacts, with reduced reproductive potential and lifespan of offspring when fathered by males, or sperm, that had experienced heatwaves. This male reproductive damage under heatwave conditions provides one potential driver behind biodiversity declines and contractions through global warming.

[1] School of Biological Sciences, University of East Anglia, Norwich NR4 7TJ, UK. [2] Institute of Zoology and Biomedical Research, Jagiellonian University, Gronostajowa 9, 30-387 Kraków, Poland. [3] School of Environmental Sciences, University of East Anglia, Norwich NR4 7TJ, UK. Correspondence and requests for materials should be addressed to M.J.G.G. (email: m.gage@uea.ac.uk)

Earth's climate is changing[1], and natural populations are responding to unnatural temperature changes by shifting ranges, declining and going extinct[2–4]. There are hundreds of studies describing concordant declines, extinctions or range shifts across a diversity of taxa in terrestrial, marine and fresh-water ecosystems that can be explained by climate change[4–6]. Despite all this, we have "disturbingly limited knowledge"[7] on the proximate causes behind these changes, and systematic experiments on specific vulnerabilities and mechanistic drivers have been prioritised to enable biodiversity predictions[8]. Here, we apply such an experimental approach to understand how climate change influences a key biological trait for population viability by investigating the detailed impacts of heatwave conditions on reproductive function in a model insect system.

With a warmer, more volatile atmosphere, extreme climatic events such as heatwaves are predicted to become more common[9,10]. Heatwaves, commonly defined as conditions when daily thermal maxima exceed the average local maximum by 5 °C for more than 5 days[11], are predicted to become longer[10,12], more intense[13,14], more frequent[9,15] and more widespread[11]. Because they generate unusually extreme thermal conditions, with often short and stochastic onsets, heatwaves are likely to be particularly disruptive for biological function[16]. Heatwaves have substantial and recognised impacts on human activity and health[17], with the 2003 summer heatwave across Europe being responsible for 70,000 deaths[18]. However, consequences for biodiversity have received far less attention, despite increasing recognition of the significance of Extreme Climatic Events for ecological systems[8,19], and some evidence of the potential for heatwaves to substantially impact biodiversity[20].

Reproductive sensitivity to increases in temperature that organisms often experience in the natural environment is well known in mammals, where adaptations that allow testicular cooling of 2 to 8 °C below core body temperature are essential to allow normal male fertility[21]. Even mild increases in the ambient thermal environment can disrupt male reproductive function in endotherms: for example, exposing male mice for 24 h to an air temperature of 32 °C resulted in fertility declines of ~75%[22], and a number of similar studies reveal such sensitivities[21,23]. By contrast with the research on endotherms, however, very limited attention has been given to 'cold blooded' taxa[24]. This is surprising, because the vast majority of biodiversity is comprised of ectothermic taxa[25], where biological functions are more directly influenced by changes in the thermal environment[26]. Reproductive sensitivity to temperature is known in *Drosophila melanogaster* with most populations becoming non-viable above 30 °C, the temperature where male reproduction ceases[24,27]. There is local adaptation to this male sensitivity, with temperate *Drosophila* populations failing to reproduce at lower thermal thresholds than tropical strains[27], but details of the causes and wider consequences of reproductive compromise in ectothermic taxa remain a 'significant but neglected phenomenon'[24]. Using a series of experiments with a model insect system, we first measure the impact of heatwave conditions on reproductive performance of males and females, then identify specifically how key traits are impacted, and finally evaluate their wider transgenerational consequences.

We performed experiments using the red flour beetle *Tribolium castaneum*, an endopterygote coleopteran with developmental and reproductive physiology representative of most insect groups, and therefore relevant to a huge number of ectotherms, many of which are under threat from climate change[26,28]. *T. castaneum* occupies tropical and warm-temperate thermal niches[29], where most terrestrial biodiversity exists[30]. We found that heatwave conditions (5 to 7 °C above the system's optimum[29] for 5 days) damaged male reproductive potential, whereas females

were largely unaffected. Heatwaves halved male male fertility, and compromised sperm competitive ability. Successive heatwaves exacerbated these effects, with a second heatwave inducing almost complete sterility in males. Inseminated sperm within female storage were also sensitive to thermal stress, reducing the female's subsequent reproductive fitness following a heatwave. Using in vivo and in vitro assays, we found that heatwaves reduced sperm number and viability, and compromised their ability to reach female storage for fertilisation. We also found transgenerational impacts of heatwaves: reproductive potential of male offspring was significantly reduced if they had been fathered by males or sperm that had previously experienced thermal stress, and offspring lifespan was shortened if fathers had experienced a heatwave. We therefore find in a model insect that male reproduction and sperm function are widely damaged by heatwave conditions, providing one explanation for how population viability could be compromised by global warming.

## Results and discussion

**Heatwave impacts on sex-specific reproductive output**. We found clear evidence that male reproduction was sensitive to thermal stress. Males exposed to a single heatwave showed a significant reduction in their subsequent ability to sire offspring, more than halving reproductive output following a 42 °C heatwave, compared with either 35 or 30 °C controls (Fig. 1; Table 1). By contrast, female reproductive output was unaffected by the same heatwave conditions (Fig. 1). We therefore reveal the characteristic male-specific sensitivity of reproductive output to thermal conditions in our insect model, as recognised in some endotherm groups[21,23], and recently in a few ectotherm species[24,31–39]. We also discovered that heatwave conditions impaired male reproductive competitiveness, reducing the number of offspring sired by second-mating males within two-male sperm competitions from 80 to 30% (Fig. 2b; Table 1). Since polyandrous mating and post-copulatory sperm competition is the standard route to fertilisation in the majority of species[40], particularly insects[41], these findings reveal a significant impact on male reproductive fitness within the relevant context of sperm competition[41]. Finally, we assessed impacts of additional heatwaves on male reproductive output, and find no evidence for short-term acclimation or 'hardening' to thermal stress so that reactions to subsequent heatwaves are better resisted. Instead, the impact of a second heatwave, 10 days after the first, was additive or even multiplicative, with males becoming almost completely sterile following a second 5-day heatwave that is 7 °C above the 35 °C optimum for population productivity in *T. castaneum* (Fig. 2c).

**Heatwave impacts on mating behaviour and fertility**. Having identified male-specific sensitivity to heatwave conditions in competitive and non-competitive contexts, we determined which mechanisms and drivers explained this loss of reproductive performance. Detailed assays of male mating behaviour revealed that heatwave conditions subsequently increased the latency before a male's first successful mating, prolonged the duration of copulation and decreased the frequency of mating (Supplementary Figure 1; Table 1). However, these males were still able to achieve an average of five copulations per female per hour through their 1-h observation period (compared with eight matings by untreated control males, Supplementary Figure 1a). Moreover, dissections of 36 females paired for 1 h with males that had been previously exposed to 42 °C heatwaves revealed that every male had successfully transferred sperm. A single mating in *T. castaneum* is sufficient for females to fertilise ~700 eggs across four months of oviposition[42], so the findings that heatwaved males

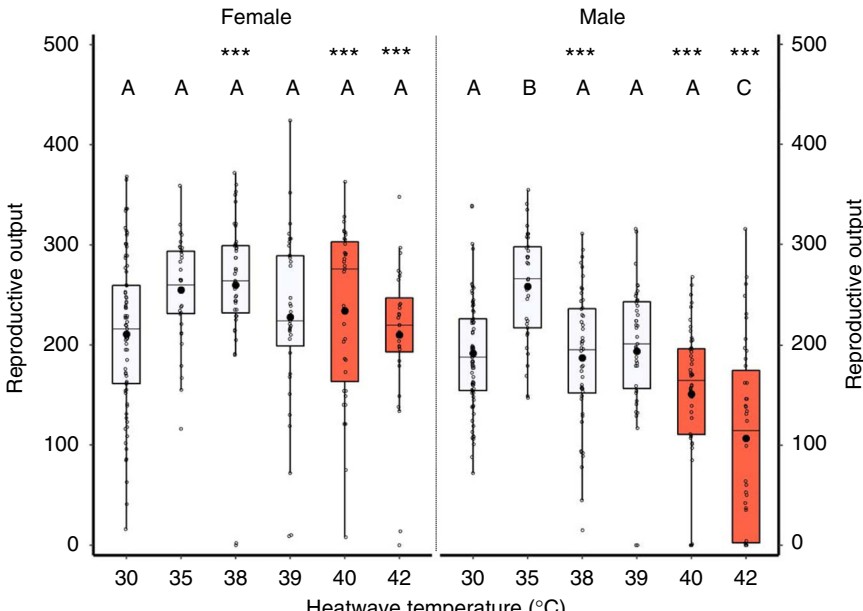

**Fig. 1** Reproductive output of males and females following exposure to 5-day heatwaves at increasing temperatures. Reproductive output is the average sum of offspring produced per breeding pair following 20 days of oviposition. Orange boxes highlight temperatures defined as heatwaves in this species. Sample sizes from left to right ($n_{females}$ = 75, 34, 43, 35, 35, 28; $n_{males}$ = 79, 33, 48, 43, 48, 42). Boxplots display a mean dot, median line, interquartile range (IQR) boxes, 1.5*IQR whiskers and data points. Significance thresholds: ***$P$ < 0.001 within temperature between sexes; letters denote differences to the 30 °C treatment within sexes. Raw data are available in the associated Source Data file.

mated an average of five times across only 1 h of observation, and all successfully transferred sperm, indicate that changes to male mating behaviour could not explain the 50% reduction in male reproductive performance following heatwave exposure (Fig. 1). Instead, we found that the reduction in reproductive output is primarily explained by male failure to stimulate female fecundity and fertilise eggs through to hatch: females mated to heatwave-exposed males reduced the number of eggs they laid by one-third (Supplementary Figure 2; Table 1), and only 40% of these eggs successfully hatched, contrasting with normal hatch rates of ~90% (Fig. 3h; Table 1). Once successfully hatched, offspring development through the larval and pupal stages was unaffected by the heatwave exposure of males (Fig. 3h).

**Heatwave impacts on sperm function.** Further experimental assays were conducted (see Methods) to identify the mechanisms explaining this decline in male fertility, and we found clear and profound impacts on the production and subsequent function of spermatozoa. Males exposed to 42 °C heatwaves showed a 75% reduction in ejaculate sperm number (Fig. 3a). Observationally, sperm masses dissected from the spermatophores deposited by heatwave-treated males also contained obvious quantities of globular detritus, while ejaculates from control males showed no such detritus and only contained sperm cells (Supplementary Figure 3). It was not possible to quantify this detritus as it was bound within the sperm mass, and broke up as the sperm mass was dispersed for counting, but its appearance and position within the spermatophore sperm mass was consistent with it being material from damaged and degraded sperm cells. These observations were further confirmed by analyses of sperm viability: in addition to the 75% reduction in sperm number, only one-third of sperm cells produced by males following heatwave conditions were alive, whereas more than 80% of sperm cells produced by control males were viable (Fig. 3c, f, g). Finally, by tracking the in vivo presence and position of Green Fluorescent Protein-labelled (GFP) sperm[43] within the female reproductive

tract, we confirmed a significant reduction in the transfer and storage of sperm by heatwave-exposed males into the bursa copulatrix and spermatheca, both of which are important sites for short-term and long-term fertilisation storage in *T. castaneum*[42,43]. The amount of GFP-labelled sperm present in these sites 24 h after mating was reduced by two-thirds when females had mated with males previously exposed to 42 °C heatwave conditions (Fig. 3b, d, e), either as a consequence of lower sperm densities, or due to reduced GFP excitation associated with dying sperm.

Although we found that females showed reproductive tolerance to thermal stress, we also discovered that inseminated sperm were sensitive to heatwaves, thereby causing a secondary loss of reproductive fitness. Experimentally mated females containing mature sperm already transferred to storage in the bursa copulatrix and/or spermatheca showed 33% declines in reproductive output following heatwave exposure, compared to age-matched females who were exposed to heatwaves before mating and sperm storage (Fig. 2a). In almost all internally fertilising ectotherm animals, representing the majority of eukaryotic biodiversity[25], females store sperm in specialised organs to allow fertilisation and reproduction to take place independent of the presence of mating males[44]. The finding that sperm are sensitive to heatwave conditions once stored within the female tract therefore has relevance for reproduction and population viability across a significant fraction of global biodiversity.

Our combined results demonstrate that male reproductive performance is specifically sensitive to heatwave conditions through thermal damage to fertility and sperm competitiveness, and that these conditions cause reductions in sperm number, migration to female storage and viability. Although female reproduction is intrinsically unaffected by exposure to the same thermal conditions, we identify population vulnerability through inseminated sperm held in female storage also being specifically sensitive to heatwave conditions. Although our experiments focus on temperature, spermatozoa are among the most complex and diverse eukaryotic cell types[45], with functional sensitivities to

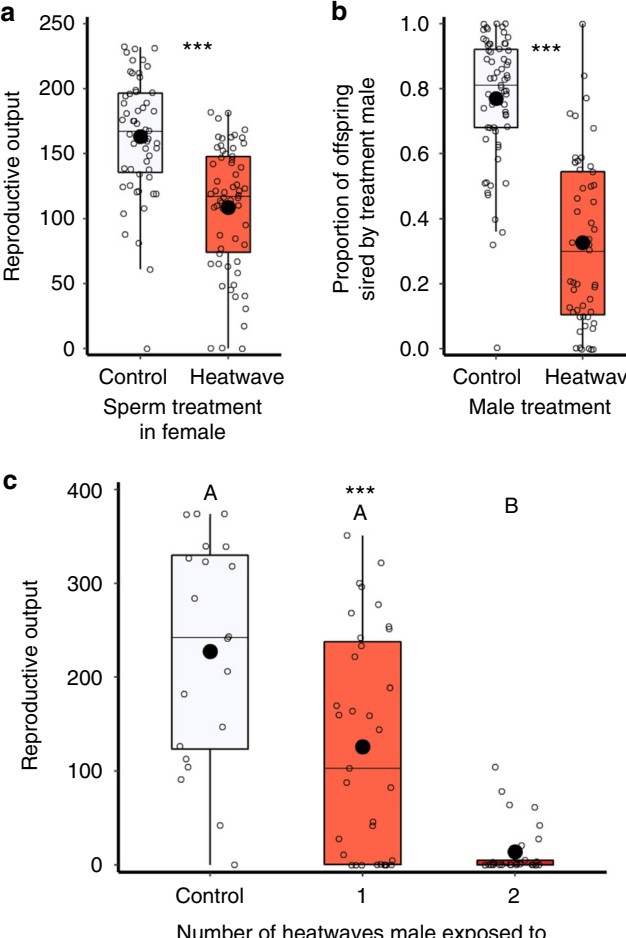

**Fig. 2** Consequences of heatwaves for male and female reproductive output. Orange boxes highlight heatwave treatments, and red box highlights heatwave exposure to both female and sperm or a second heatwave. **a** Impacts of heatwaves on inseminated sperm: reproductive output of females exposed to heatwaves before mating and sperm storage (control: $n = 55$) compared to females exposed to heatwaves after mating and with inseminated sperm in storage (heatwave: $n = 62$). Reproductive output is number of offspring produced by breeding pairs following 10 days of oviposition. **b** Heatwave impacts on sperm competitiveness indicated by proportions of offspring sired by control ($n = 65$) versus heatwave-treated ($n = 51$) males with females previously mated to single rival control marker males. **c** Impacts of additional heatwaves on male reproductive output across 20 days of oviposition: control males (white, $n = 20$), single heatwave (orange, $n = 35$) and double heatwaves (dark red, $n = 29$). Boxplots display a mean dot, median line, IQR boxes, 1.5*IQR whiskers and data points. Significance thresholds: ***$P < 0.001$, with letters identify significant differences between groups. Raw data are available in the associated Source Data file

physiological[46] and genetic[47] stress, so our findings may also be due to a general spermatozoal susceptibility to stress. In addition, we exposed adult beetles to temperature increases for 5 days in order to replicate heatwave conditions, however, recent work has shown that thermal impacts on reproduction can also occur over relatively short windows of more acute exposure[48,49]. Whichever of these situations apply, our combined findings could shed light on why populations have declined as a result of increased thermal or general stress from climate change[2–4,6,7,19,20].

**Transgenerational impacts of heatwaves**. In addition to these direct effects, we also discover a less noticeable, longer-term

impact of heatwaves: transgenerational damage. Using two-generation experiments (Methods), we found that the adult life-span of offspring sired by males that had previously experienced 40 °C heatwaves was reduced significantly compared to the longevity of offspring sired by controls (Fig. 4a, b; Table 1). Similarly, we found that the reproductive potential of sons fathered by heatwave-exposed males was reduced when they were given the opportunity to mate with multiple females (Fig. 4c, d, e, f). Sons of males that had been exposed to a single heatwave in the previous generation showed a 25% reduction in mating success and subsequent offspring production, compared with controls whose fathers had not experienced heatwave conditions (Fig. 4c, d). Critically, these same transgenerational effects were also evident when mature inseminated sperm alone had experienced the same conditions within the female: sons fertilised by heatwave-exposed spermatozoa in the female tract suffered 25 to 40% reductions in their reproductive output and mating success, compared with sons from parents where the mother had experienced the heatwave before mating and sperm storage, or controls where no heatwaves were experienced (Fig. 4e, f).

Transgenerational fitness damage is known to occur in a range of species as a consequence of stressors such as irradiation[50], toxic chemicals[51], sensory perturbations[52] and ageing[53]. This damage has also been found to compromise male reproductive function[54,55]. In a recent study exposing field crickets (*Gryllus bimaculatus*) to 24 and 28 °C regimes, the warmer treatment when exposed to adult males was found to reduce ejaculate sperm number (with the reverse effect seen when 28 °C exposure took place through the pre-adult stages as well)[39]. Adult males exposed to the warmer 28 °C regime also fathered offspring that exhibited reduced survival (and again the reverse effect was seen with improved offspring survival if warmer 28 °C exposure occurred throughout development)[39]. In this study, we believe we present the first evidence for significant negative transgenerational effects as a consequence of heatwave exposure in the parental generation specifically through thermal impacts on mature, inseminated spermatozoa stored within the female reproductive tract. Such damage could occur through physical damage to the paternal haplotype within the sperm nucleus, possibly as a result of thermal impacts on DNA fragmentation and mutations[56,57]. Oocyte and zygote cell repair mechanisms can reverse sperm DNA damage through embryogenesis[58], but this may not be possible following our heatwave treatment conditions if the DNA damage is sufficiently severe. Alternatively, epigenetic alterations to gene expression following heatwave conditions may arise through changes to chromatin condensation[59,60], methylation[61] and/or non-coding RNA transfer[62]. We urge future research into (1) the molecular basis of this transgenerational heatwave damage, (2) whether females have evolved mate choice strategies to avoid male-derived thermo-sensitive infertility and (3) for the consequences of our findings of heatwave damage to male reproductive function to be examined in a broader range of taxa.

## Methods
**Stock culture maintenance**. The red flour beetle *Tribolium castaneum* is a tractable research model for studying reproduction[29,64,65]. We used the outbred 'Kraków Super Strain' (KSS) created in 2008 by combining 35–60 individuals from 11 different strains to promote genetic diversity[66]. Stocks were maintained under standard conditions (30 ± 1 °C, 60 ± 5% RH and 16L: 8D photoperiod) in ad libitum fodder consisting of organic flour and yeast (9:1 by volume) topped with oats for traction[65]. Populations were maintained as non-overlapping generations, renewed every 35 days by transferring ~300 sexually mature adults to fresh fodder for 7 days mating and oviposition, then removing adults to allow egg and larval development. Unless otherwise stated, all individuals used in experiments were sexed as pupae, kept in single-sex groups of 20 individuals in 5 cm petri dishes to eclosion and sexual maturity at 12 ± 2 days, then randomly assigned to treatments. During maturation, one sex was identified with a dot on the dorsal thorax using correction fluid (Tippex, France). This marking method has no significant effect on

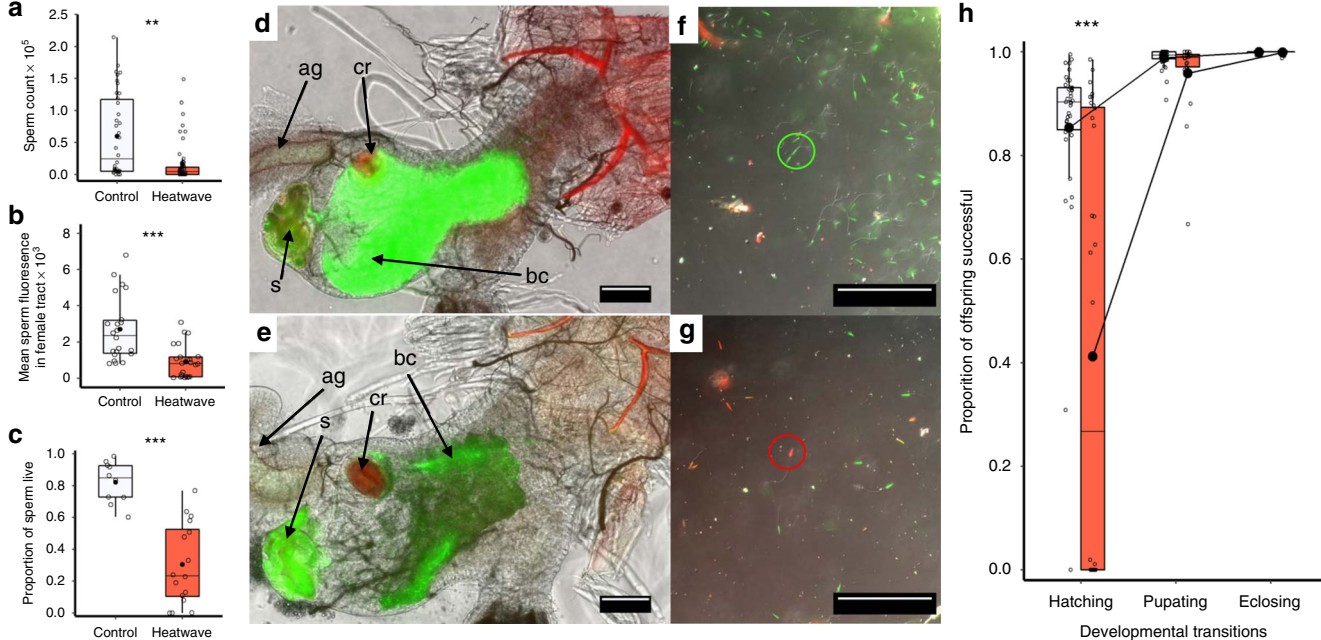

**Fig. 3** Impact of heatwaves on sperm function and male fertility. Orange boxes indicate heatwave treatment. **a** Ejaculate sperm number of control ($n = 38$) and heatwave-treated ($n = 56$) males. **b** Sperm-derived fluorescence measures across female reproductive tracts following matings with control ($n = 22$) or heatwave-treated ($n = 24$) green fluorescent protein (GFP) males. **c** Ejaculate sperm viability of control ($n = 10$) and heatwave-treated ($n = 16$) males. **d**, **e** Composite brightfield and fluorescence (green and red spectrum; see Supplementary Figure 7 for autofluorescence control protocols) images of female reproductive tracts following mating to control (**d**) versus heatwave-treated (**e**) GFP males. Identified structures are: (ag) accessory gland, (cr) chitin ring, (bc) bursa copulatrix, (s) spermatheca. **f**, **g** Composite DIC and fluorescence images of dispersed ejaculates from control (**f**) and heatwave-treated (**g**) males following sperm viability staining where live sperm heads take up the green stain (green circle) and dead sperm take up the red stain (red circle). **h** The proportion of offspring sired by control (white) or heatwave-treated (orange) males transitioning between each juvenile life stage, indicating the primary impact on egg development and/or hatch. Sample sizes from left to right ($n = 39, 32, 38, 18, 38, 18, 39, 32$). Boxplots display a mean dot, median line, IQR boxes, 1.5*IQR whiskers and data points. Significance thresholds: ***$P < 0.001$; *$P < 0.05$, with letters identifying significant differences between groups. Raw data are available in the associated Source Data file.

reproductive output across 20 days of oviposition (marked versus unmarked females $\chi^2_{(1,36)} = 0.7$, $P = 0.407$; $z = -0.8$, $P = 0.407$, $n = 19 + 19$).

**Heatwave conditions.** Heatwave treatments exposed individuals for 5 days to temperatures that exceeded the optimum by 5 °C, corresponding with the common definition of a heatwave event[14]. The optimum temperature for population productivity in *T. castaneum* is 35 °C[29,67], which our assays confirmed (Fig. 1). Experimental heatwaves therefore exposed individuals to temperatures of 40 to 42 ± 1 °C. These conditions have been recorded in the natural environment across more than 90 countries[68]. Heatwave conditions were applied using Octagon 20 incubators (Brinsea Ltd, UK), and the humidity of all treatments was maintained at 60 ± 5% RH. Beetles were exposed to heatwaves in single-sex groups of 20 individuals in 5 cm petri dishes containing standard fodder and positioned in the central plane of the incubator. Temperatures did not exceed 1 °C above or below the treatment set point, checked using a 35–45 °C mercury incubation thermometer (G.H. Zeal Ltd, Zeal House, 8 Deer Park Road, London, SW19 3UU, U.K.) calibrated to United Kingdom Accredited Service standards (Charnwood Instrumentation Services Ltd, 81 Park Road, Coalville, Leicestershire, LE67 3AF, UK). Following treatments, all individuals experienced 30 ± 1 °C for 24 h, before running reproductive output assays at 30 ± 1 °C.

**Reproductive output: heatwave impacts on adults.** Supplementary Figure 4a presents these experimental protocols. Reproductively mature males and females were exposed to 5-day thermal treatments at 30 °C ($n_{Males} = 79$, $n_{Females} = 75$), 35 °C ($n_M = 33$, $n_F = 34$), 38 °C ($n_M = 48$, $n_F = 43$), 39 °C ($n_M = 43$, $n_F = 35$), 40 °C ($n_M = 48$, $n_F = 35$), or 42 °C ($n_M = 42$, $n_F = 28$). After treatment, and a further 24 h at 30 °C, they were monogamously paired with untreated mates for 2 days at 30 °C in 4 ml vials containing 0.5 g flour and yeast topped with oats. Following mating, males were removed and females isolated in 5 cm petri dishes for oviposition into 7 g flour and yeast with 3 g of oats on the surface for 20 days at 30 °C, using two separate 10-day blocks to reduce overlapping generations (Supplementary Figure 4a). After removing the female at day 20, eggs and larvae produced over this period were left to develop in standard conditions at 30 °C for 35 days until they emerged to be counted as mature adults. Reproductive output of each breeding pair was therefore the number of offspring successfully produced over 20 days of

oviposition, which correlates significantly with lifetime output and accounts for ~50% of a female's total potential reproductive output under similar conditions across 150 days of oviposition[66].

**Reproductive output: heatwave impacts on sperm in females.** Supplementary Figure 4b presents these experimental protocols. Impacts on individual spermatozoa were measured by exposing sperm stored within the reproductive tract of mated females to heatwave conditions, comparing against females which received the same heatwave treatment but immediately prior to mating and sperm storage (Supplementary Figure 4b). Thus, females were either mated, then exposed to heatwaves ($n = 62$); or exposed to heatwaves, then mated ($n = 55$). Following either treatment, females were transferred to 5 cm petri dishes for oviposition across three separate 5-day blocks under standard conditions, counting the number of offspring produced after 35 days of development. Five-day blocks were applied so that we could control for any differential sperm ageing effects that may have occurred between insemination and the period of reproductive fitness measurement: reproductive output by females in the 'sperm + female heated' treatment was compared across the first 10 days of oviposition following treatment (and therefore 5 days following the timing of insemination), whereas output in the 'unmated female heated' was compared following oviposition from day 5 to 15 (again, 5 days following insemination). We also ran comparisons of reproductive output for all 15 days of oviposition. Both comparisons showed significant 26 to 31% declines in female reproductive output when females had been exposed to heatwave conditions containing sperm in storage. Results controlling for sperm age and comparing reproductive output across 10 days of oviposition are in the main document and Fig. 2a. Comparisons of the total 15 days of reproductive output yielded similar results with significant declines in reproductive output when sperm had experienced heatwave conditions within female storage ($\chi^2_{(1,115)} = 17.1$, $P < 0.001$; $z = -4.1$, $P < 0.001$).

**Reproductive output: heatwave impacts on sperm competition.** Supplementary Figure 5 presents these experimental protocols. To assess impacts within the relevant context of sperm competition, we measured how heatwave conditions influenced a male's subsequent ability to win fertilisations within females that had previously been mated to untreated, marker males. Males were sexed as pupae, and

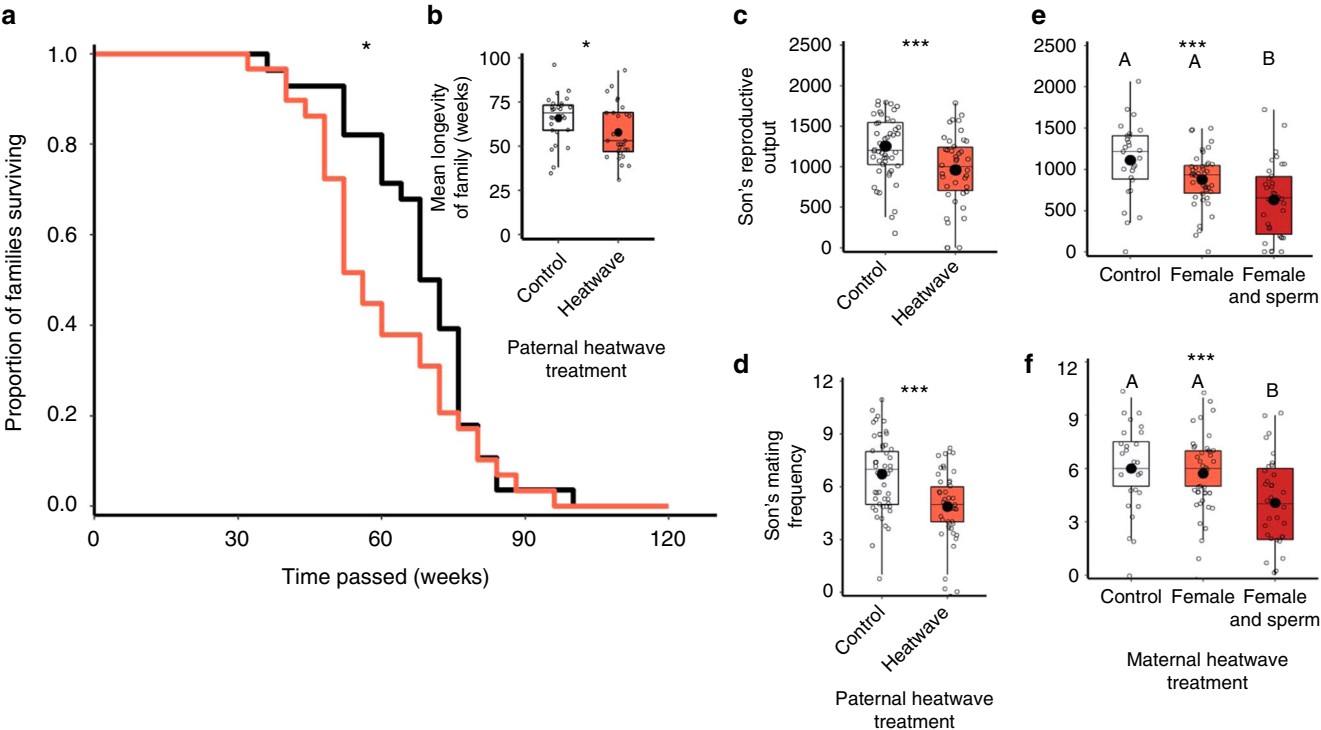

**Fig. 4** Transgenerational effects of heatwaves on offspring fitness. Orange markers indicate heatwave treatments, red markers indicate heatwave treatments of both female and sperm. **a** Survival curves of randomly-sexed adult offspring from either control (black, $n = 28$) or heatwave-treated (red, $n = 29$) fathers, with insert boxplot (**b**) of the adult offspring lifespans. Offspring were kept isolated as single unmated adults, with fodder renewed monthly for up to two years. Each data point represents the mean of a family consisting of four sibling replicates. **c** Total reproductive success across 20 days of oviposition of sons from control ($n = 48$) versus heatwave-treated ($n = 42$) fathers given mating opportunities across a series of 13 unmated females. **e** Total reproductive success across 20 days of oviposition of sons from control ($n = 27$) and heatwave-treated, unmated mothers ($n = 42$) and mated, heatwave-treated mothers carrying inseminated sperm in storage ($n = 34$); reproductive output of sons measured following mating opportunities across a series of 13 unmated females. **d**, **f** Successful mating frequencies of sons when given mating opportunities across a series of 13 unmated females, depending on whether their fathers had been exposed to heatwaves (**d**), or whether mothers or inseminated sperm within mothers had been exposed to heatwaves (**e**). Protocol, sample sizes and treatments match **c** and **e**. Boxplots display a mean dot, median line, IQR boxes, 1.5*IQR whiskers and data points. Significance thresholds: ***$P < 0.001$; *$P < 0.05$, with letters identify significant differences between groups. Raw data are available in the associated Source Data file.

then isolated from eclosion until experimental mating to standardise and prevent any confounds from uncontrolled same-sex behaviour activity[69]. Treatment males were exposed to 5 days at 30 °C (controls) or 42 °C (heatwaves), followed by 24 h at 30 °C. During this 24 h period, control females were mated to 'Reindeer' marker males. The Reindeer (*Rd*) mutation for clubbed antennae is dominant and maintained homozygous in our stock. Offspring sired by Reindeer males will inherit the clubbed antennae phenotype, whereas offspring sired by the wild type males develop normal filiform antennae, allowing paternity to be assigned treatment group males[70]. After 24 h mating with *Rd* males, females were then mated with either control ($n = 65$) or heatwave-treated ($n = 51$) males for 24 h, and then transferred to oviposit individually in 5 cm petri dishes for 7 days. Following oviposition, offspring were left to develop for 35 days, after which the relative numbers of wild type and *Rd* offspring were counted to measure differences in paternity and relative sperm competitiveness between the heatwave and control male treatment groups.

**Reproductive output: double heatwave impacts**. Supplementary Figure 6 presents these experimental protocols. To measure the impact of additional heatwaves, adult males were exposed to three treatments: (1) Control: 5 days of exposure to 30 °C ($n = 20$); (2) Single heatwave: 5 days of heatwave exposure at 42 °C ($n = 35$); and (3) Double heatwaves: 5 days of heatwave exposure at 42 °C followed by 10 days at 30 °C followed by a second 5 days of heatwave exposure at 42 °C ($n = 29$). Following each treatment, males were maintained for 24 h at 30 °C before being monogamously paired to untreated adult mature females for 2 days in 4 ml vials, after which females were transferred individually to 5 cm petri dishes for 20 days of oviposition in standard conditions, across two 10-day blocks. After 20 days, females were removed and all offspring allowed to develop for 35 days so that offspring production could be counted. To minimise developmental effects through initial spermatogenesis, all males were reproductively mature ($12 \pm 2$ days post eclosion) and received their initial 5-day treatments simultaneously, with males in group 3 experiencing their second heatwave at age $27 \pm 2$ days post

eclosion. Thus, all males were reproductively mature when exposed to single or double heatwaves (Supplementary Figure 6).

**Heatwave impacts on male mating behaviour**. Males sexed as pupae were individually isolated before their mating behaviour assay to prevent any same-sex activity and to standardise all individuals prior to each trial[64,69]. At adult maturity, males were exposed for 5-day treatments at 30 °C ($n = 25$), 39 °C ($n = 24$), 40 °C ($n = 21$), 41 °C ($n = 24$) or 42 °C ($n = 14$), followed by 24 h at 30 °C (Supplementary Figure 4c). Following treatment, males were paired with untreated control females at 30 °C in 1 cm² mating arenas for 1 h, and all mating activity video-recorded using Sony digital video cameras. Replaying the 1-h film sequence for each pair, we recorded: (1) the period of latency to first mating, (2) the total number of matings and (3) the duration of each mating. Matings were defined when the pair achieved unbroken mounting and copulatory contact for more than 35 s, which is the average minimum time for successful spermatophore transfer in *T. castaneum*[71].

To assess the probability of subsequent spermatophore transfer in matings by males previously experiencing heatwave conditions, we ran an additional assay in which males ($n = 36$) that had previously received a 5-day 42 °C treatment were paired monogamously with untreated females for 1 h in 1 cm² mating arenas, after which females were frozen at −20 °C, before being dissected to check for successful sperm transfer.

**Impacts on fertility, fecundity and offspring development**. Supplementary Figure 4e presents these experimental protocols. To determine whether the decline in male reproductive fitness following heatwave exposure was a consequence of (1) reduced egg hatch (fertility), (2) reduced numbers of eggs produced (fecundity), or impacts on offspring development through the (3) larval and 4) pupal stages, we ran breeding assays to measure separate impacts on each (Supplementary Figure 4e). Males exposed to either 30 °C control or 42 °C heatwave conditions followed by 24 h at 30 °C were then paired monogamously with untreated and

| Experiment | Fixed factor | DF | $\chi^2$/F | P | Model, error distribution and link function | $R^2$ [63] |
|---|---|---|---|---|---|---|
| **Table 1 Model summaries for effects of heatwave exposure on reproductive output** | | | | | | |
| Male and female reproductive fitness Fig. 1 | Heatwave temperature × sex | 5 | 40.2 | <0.001 | GLM Quasi-Poisson (log) | 21% |
| | Heatwave temperature | 5 | 71.7 | <0.001 | | |
| | Sex | 1 | 51.2 | <0.001 | | |
| | Residual | 532 | | | | |
| Stored sperm reproductive fitness Fig. 2a | Female stored sperm heatwave treatment | 1 | 14.1 | <0.001 | GLM Negative binomial (log) | 10% |
| | Residual | 116 | | | | |
| Paternity proportion in competition Fig. 2b | Male heatwave treatment | 1 | 97.3 | <0.001 | GLM Quasi-binomial (logit) | 44% |
| | Residual | 115 | | | | |
| Impacts of a second heatwave on male reproductive fitness Fig. 2c | Male heatwave number | 2 | 25.2 | <0.001 | GLM Negative binomial (log) | 21% |
| | Residual | 82 | | | | |
| Sperm count Fig. 3a | Male heatwave treatment | 1 | 7.2 | 0.007 | GLM Negative binomial (log) | 6% |
| | Residual | 93 | | | | |
| Sperm distribution in female storage Fig. 3b, d, e | Male heatwave treatment | 1 | F = 19.3 | <0.001 | GLM Gaussian (log) | 31% |
| | Residual | 47 | | | | |
| Sperm viability Fig. 3c, f, g | Male heatwave treatment | 1 | 10.1 | <0.001 | GLM Binomial (logit) | 27% |
| | Residual | 24 | | | | |
| Egg hatch Fig. 3h | Male heatwave treatment | 1 | 17.9 | <0.001 | GLM Quasi-binomial (logit) | 20% |
| | Residual | 70 | | | | |
| Larval development Fig. 3h | Male heatwave treatment | 1 | 3.1 | 0.074 | GLM Quasi-binomial (logit) | 8% |
| | Residual | 55 | | | | |
| Pupal eclosion Fig. 3h | Male heatwave treatment | 1 | 0.9 | 0.334 | GLM Binomial (logit) | 3% |
| | Residual | 55 | | | | |
| Egg to adult success Fig. 3h | Male heatwave treatment | 1 | 19.3 | <0.001 | GLM Quasi-binomial (logit) | 21% |
| | Residual | 70 | | | | |
| Offspring longevity Fig. 4a, b | Paternal heatwave treatment | 1 | 4.7 | 0.030 | Accelerated Failure Time Survival Model Gaussian | |
| | Residual | 56 | | | | |
| Reproductive fitness of sons Fig. 4c | Paternal heatwave treatment | 2 | 9.7 | <0.001 | GLMM Gaussian (identity) | 10% |
| | Residual | 88 | | | | |
| Mating success by sons Fig. 4d | Paternal heatwave treatment | 2 | 13.2 | <0.001 | GLMM Poisson (log) | 14% |
| | Residual | 88 | | | | |
| Reproductive fitness of sons Fig. 4e | Female stored sperm heatwave treatment | 2 | 19.8 | <0.001 | GLM Quasi-Poisson (log) | 14% |
| | Residual | 101 | | | | |
| Mating success by sons Fig. 4f | Female stored sperm heatwave treatment | 2 | 14.4 | <0.001 | GLM Poisson (log) | 10% |
| | Residual | 101 | | | | |
| Mating frequency Suppl Figure 1a | Male heatwave treatment | 4 | 36.9 | <0.001 | GLM Poisson (log) | 16% |
| | Residual | 104 | | | | |
| Mating duration Suppl Figure 1a | Male heatwave treatment | 4 | F = 14.3 | <0.001 | GLM Gamma (identity) | 7% |
| | Residual | 717 | | | | |
| Female fecundity Suppl Figure 2 | Male heatwave treatment | 1 | 15.2 | <0.001 | Negative binomial (log) | 9% |
| | Residual | 134 | | | | |

unmated females in 0.5 g flour and yeast topped with oats for 2 days at 30 °C. After mating, females were transferred to individual 4 ml vials with 0.5 g of pre-sieved flour and yeast topped with oats for oviposition under standard conditions. Every 2 days (and therefore before egg hatch) through a 10-day oviposition period, females were transferred to new vials, and eggs in the fodder sieved out using 300 μm mesh (Endecotts Ltd., London, UK). Separated eggs were dispersed on black tiles using a fine paintbrush and counted under a Zeiss Stemi 2000-C stereomicroscope at ×10 magnification to give a fecundity measurement for control ($n = 59$) and heatwave treatments ($n = 76$). For a random subset of the control ($n = 40$) and heatwave ($n = 40$) treatments, all eggs were transferred immediately to 5 cm petri dishes containing 7 g of pre-sieved fodder to allow development. Ten days later, after which all successfully fertilised eggs would have hatched (egg development to hatch takes ~4 days under standard conditions in *T. castaneum*[29], early stage larvae were sieved again from the fodder within each 2-day oviposition block and counted to provide egg hatch scores, before being returned to fodder. Twenty days later, pupae were counted in each block to quantify successful larval development and, at 35 days, when all hatched eggs, larvae and pupae would have developed to successful eclosion, adult offspring were counted.

**Heatwave impacts on ejaculate sperm counts.** Mature males were exposed to 5-day treatments of either 30 °C control ($n = 36$) or 42 °C heatwave ($n = 56$)

conditions, then paired with a series of five mature untreated and unmated females in 1 cm$^2$ mating arenas. Each male was paired with a female for 15 min, before being transferred to the next female. Access to a series of females allowed us to measure the rate of successful sperm transfer, and increased the probability that a male would transfer at least one spermatophore successfully to allow sperm counting (Supplementary Figure 4d). Immediately following each 15-min mating period, females were frozen at −20 °C for subsequent dissection and sperm count. Females were dissected in saline buffer (1% NaCl solution) under a Zeiss Discovery V.12 stereomicroscope (Carl Zeiss, Jena, Germany) under ×20 magnification. Using fine forceps, the female tract was removed, the bursa copulatrix cut open, and the tract then separated from any spermatophore which was isolated in 100 μl of saline buffer on a cavity slide. The spermatophore was then broken apart using size 0 dissection pins and the sperm mass released and dispersed into the buffer, before being washed off the slide and into a 10 ml tube using 3 ml of distilled water expelled from an autopipette. Each solution was then gently mixed before taking three 20 μl subsamples which were placed on flat glass slides to dry as smears. After air-drying, the slides were dipped gently into distilled water to remove any desiccant, and re-dried. Sperm cells (including their component parts, see below) adhere to the glass and were counted within each smear using dark field phase-contrast microscopy at ×200 magnification on an Olympus BX41 microscope (Olympus Corporation, Tokyo, Japan)[72]. Because many sperm cells had suffered membrane disruption and separation into their two elongate mitochondrial

derivatives, possibly due to freeze damage, sperm number in each smear was determined by counting the total number of mitochondrial derivatives divided by two, added to the total number of undamaged sperm cells in each smear. The average sperm count for the three smears was then multiplied by their dilution factor (×155) to calculate total spermatophore sperm count.

**Heatwave impacts on sperm migration in the female tract**. Heatwave impacts on sperm function and distribution following insemination were assayed using males from a *T. castaneum* strain modified to incorporate a green fluorescent protein (GFP) into sperm chromatin[43], enabling imaging of sperm distribution within the semi-transparent female reproductive tract (Fig. 3). Before mating, mature GFP males were exposed to 5-day treatments of either 30 °C control ($n = 22$) or 41 °C heatwave conditions ($n = 24$), followed by 24 h at 30 °C. Following treatment, GFP males were paired with mature untreated and standard KSS females for 90 mins. Following insemination, and to allow sperm to exit the spermatophore completely and reach longer-term storage in the bursa copulatrix and spermatheca[42,43,64], females were snap-frozen 24 h after mating at -80 °C. The intact reproductive tracts of these females were then removed through micro-dissection of defrosted specimens under a Zeiss Discovery V.12 stereomicroscope (Carl Zeiss, Jena, Germany) in Grace's insect buffer (Thermo Fisher, Massachusetts, USA). Following removal of the complete tract, the ovaries were separated from the upper tract, and the lower tract then excised from the oviduct's junction with the ovipositor, keeping the main tract containing the bursa copulatrix, spermatheca and any sperm intact. This tract was then placed in 30 μl of Grace's buffer on a slide and sealed under a 20 × 20 mm coverslip with impermeable instant contact adhesive (EVO-STIK, UK), before imaging using Zeiss Axiocam and Axiovision hardware and software.

Supplementary Figure 4d and 7 present these protocols. To visualise fluorescing sperm, brightfield and fluorescence images were acquired through a Zeiss ×10, 0.3 NA Plan-Neofluar objective on an AxioPlan 2ie microscope and captured with an Axiocam HRm CCD camera and Axiovision 4.8.2 software. Greater resolution of the smaller spermatheca was achieved through a Zeiss ×20, 0.6 NA Plan-Apochromat objective. GFP fluorescence, primarily from sperm, was excited through a 472 ± 15 nm excitation filter, and emitted fluorescence collected through a 520 ± 17.5 nm emission filter. General autofluorescence (AF) was excited through a 562 ± 20 nm excitation filter, and the emitted fluorescence collected using a 624 ± 20 nm filter. Exposure times were kept constant between samples. Images (14-bit greyscale) of the female tract and stored sperm were analysed using a custom-written macro in Fiji (ImageJ, ver. 1.49k)[73] (Supplementary Figure 7). The macro subtracted background in each channel image using a rolling ball radius of 25 pixels for the smoothing algorithm[74]. To remove autofluorescence from the GFP-channel image so that only GFP sperm fluorescence was visible[75], the macro corrected each GFP-channel image as follows: a region of interest (ROI) was created manually in the AF-channel in an area of the image displaying high fluorescence but no corresponding fluorescence in the GFP-channel image, the mean intensity was then measured in this ROI ($Int_{Auto}$). The typical structure for this ROI was the chitinous ring at the base of the spermathecal duct (Supplementary Figure 7c). The same ROI was then applied to the GFP-channel image and the mean intensity measured ($Int_{GFP}$). A correction factor (CF) was determined by dividing $Int_{GFP}$ by $Int_{Auto}$. The AF-channel image was multiplied by CF and the resultant corrected AF image subtracted from the GFP-channel image, leaving only GFP sperm-derived fluorescence for measurement (Supplementary Figure 7d). The brightfield image was then used to define the ROI to be analysed by manually drawing around each tract's perimeter walls (Supplementary Figure 7a, d). The mean pixel intensity within this ROI was then determined, providing a measure of the presence and distribution of GFP sperm in each tract.

**Heatwave impacts on sperm viability**. The impacts of heatwave conditions on mature sperm viability were measured from spermatophores transferred at mating to control females following exposures of mature males for 5 days at either 42 or 30 °C, and 24 h at 30 °C for both groups (Supplementary Figure 4d). Because males exposed to heatwaves can take longer to mate (Supplementary Figure 1), 42 °C heatwaved males were paired with untreated and unmated females for 210 min ($n = 16$) before dissection, and 30 °C control males for 90 mins ($n = 10$). Females were dissected immediately after their pairing period, with the protocol following that for sperm counts, apart from modifications for sperm viability staining and visualisation. Once spermatophores had been separated from the female bursa copulatrix, they were held in 30 μl of Grace's insect buffer (Thermo Fisher, Massachusetts, USA) on a cavity slide. Having gently dispersed the sperm mass with size 0 dissection pins, sperm cells were stained with 2 μl of a 15-fold dilution of 2.4 mM propidium iodide and 2 μl of a 10-fold dilution 1 mM SYBER-14 dye from the LIVE/DEAD Sperm Viability Kit L-7011 (Molecular Probes, Oregon, USA). The sperm solutions were then sealed within the slide cavity using a 20 × 20 mm coverslip, and incubated for 5 mins at 27 ± 2 °C to allow stain uptake. Following incubation, image analysis took place using Zeiss Axiocam and Axiovision hardware and software. Sperm heads were imaged in (1) red and (2) green fluorescence channels, and (3) Differential Interference Contrast (for detecting non-stained sperm). All sperm observed in the viability assay took up the stain to fluoresce either red or green (Fig. 3).

Following staining and incubation using the LIVE/DEAD Sperm Viability Kit L-7011, differential-interference contrast and fluorescence images were acquired using a Zeiss ×20, 0.6 NA Plan-Apochromat objective on a AxioPlan 2ie microscope at ×200 magnification. Within 60 min of dissection, six images were captured at randomly selected locations across each diluted, incubated and stained sperm sample using a Axiocam HRm CCD camera. Propidium iodide fluorescence was excited using a 562 ± 20 nm excitation filter, and the emitted fluorescence collected with a 624 ± 20 nm filter. SYBER-14 fluorescence was excited with a 472 ± 15 nm excitation filter, and the emitted fluorescence collected through a 520 ± 17.5 nm emission filter. Using the L-7011 Sperm Viability Kit (Molecular Probes, Oregon, USA), live sperm with intact membranes take up the by SYBER-14 stain and their heads fluoresce green, while dead cells take up propidium iodide and fluoresce red. The proportion of viable sperm in each sample was calculated as the average (across the six subsamples) total number of live sperm, divided by the average total number of live sperm plus average total number of dead sperm. Counts were manual and based on colour dyed heads. Sperm survival has been previously shown to correlate with the number present[76] therefore, sperm count was included as a random factor in a Generalised Linear Mixed Model[77] (see Data Analysis).

**Transgenerational impacts of heatwaves**. Supplementary Figure 4f presents these experimental protocols. Consequences of heatwave conditions for the reproductive performance and lifespan of adult offspring in the next generation were measured following thermal exposure to males (sires), females (dams) and inseminated sperm held in female storage. Two assays were conducted to assess transgenerational heatwave effects on (1) offspring adult lifespan in both sexes, and (2) male offspring reproductive performance. Offspring mortality rates and lifespan were compared between adult offspring groups that had either been sired by males previously exposed to a 5-day heatwave at 40 °C ($n = 28$), or by control males exposed to 5 days at 30 °C ($n = 29$) (both groups held for 24 h at 30 °C before mating). Protocols to generate offspring followed those to measure reproductive fitness, after which adults were isolated individually in 4 ml vials with 0.5 g flour and yeast topped with oats under standard conditions at 30 °C. Mortalities were recorded and fodder refreshed every month for up to two years, after which all adult offspring had died. Lifespan was therefore measured in non-competitive and non-reproductive conditions, without adult interaction and with ad libitum food, providing a fair measure of intrinsic mortality in the absence of social, mating and environmental pressures. For each adult cross (40 °C heatwave $n = 28$ and 30 °C control $n = 29$), four adult offspring were randomly assigned and measured in the lifespan assay. Previous measures showed that sex ratios within offspring groups sired by males previous exposed to heatwave conditions did not depart from unity: average % male across $n = 17$ offspring groups = 51% (±2.36); Wilcoxon test of male proportion versus 0.5: $V_{17} = 79$; $P = 0.59$.

In the second transgenerational fitness assay, we measured impacts of heatwaves in the previous generation on the reproductive performance of $F_1$ male offspring. Parental adults were either exposed to 42 °C heatwaves for 5 days followed by 24 h at 30 °C, or as 30 °C controls throughout. These control and heatwave treatments were exposed to both male and female adults to assess transgenerational effects upon male offspring reproductive fitness. Male (sire) effects were measured following exposure to 30 °C control ($n = 42$) and 42 °C ($n = 48$) heatwave conditions. Female (dam) and sperm-in-storage effects (dam + sperm) were measured following exposure to: (i) 30 °C control conditions in unmated females (dam alone control, $n = 27$), (ii) 42 °C heatwave conditions for unmated females (dam alone heatwave effect, $n = 42$), and (iii) 42 °C heatwave conditions for mated females carrying sperm in storage (dam plus sperm heatwave effect, $n = 34$). Following treatment, offspring were generated as in the reproductive output assays (Supplementary Figure 4a), and individual sons isolated at the pupal stage within those pairs producing offspring for subsequent assay. A single son was assayed from each of the parental crosses to standardise family effects. Because male *T. castaneum* have high reproductive potential[65], we compared between treatment groups using an assay in which reproductive performance of individual males was measured following opportunities to mate with a series of 13 control unmated mature females, each provided to the male in 1 cm² mating arenas for 30 min. After each 30-min access period, females were removed and exchanged for a new unmated female. Males were therefore tested for their ability to mate with and fertilise up to 13 females across a 6.5 h mating trial. Following each 30-min mating opportunity, females were transferred to 5 cm petri dishes for oviposition into 7 g flour and yeast, and 3 g of surface oats in standard 30 °C conditions across two 10-day blocks, as in the reproductive fitness assay (Supplementary Figure 4a). After oviposition, eggs were left to develop in standard conditions for 35 days, after which the total number of adult offspring produced, and the number of successful matings (evidenced by some offspring production), were counted. Our two scores of individual male reproductive performance were therefore: (1) the total number females successfully inseminated across the sequence of 13, and (2) the total number of offspring sired across the 6.5 h mating trial.

**Data analysis**. Data were analysed using R 3.3.2[78], using the RStudio.0.99.903 wrapper[79]. Graphs were produced using 'ggplot{ggplot2}'[80] package within R.

Descriptive statistics (mean ± S.E.) were calculated by 'describeBy{psych}'[81]. Exploratory analysis included distribution plotting and conservative non-parametric testing on ranks prior to fitting generalised linear models (GLMs) with 'glm{stats}'[82]. Heatwave treatments were entered into analyses as fixed factors. Where sampling structure variables (=blocks or experimental repeats) were present, either group averages were calculated, or generalised linear mixed models (GLMMs) were fitted[83], using 'glmer{lme4}'[84]. Cases where individuals died midway through assays were excluded.

The most appropriate error distribution for each GLM(M) was selected by examining diagnostic residual plots[66,83,85] using 'Plot{graphics}'[86] and 'mcp.fnc {LMERConvenienceFunctions}'[87]. Count response variables, which included all experiments measuring reproductive fitness, sperm counts, fecundity and number of mating events, were initially analysed using a Poisson distribution with a log link function. Model fits were checked and over-dispersion, where the variance exceeds the mean, was assessed in GLMs using by 'dispersiontest{AER}'[88], and in GLMMs using an over-dispersion function[66]. Where over-dispersion was present, usually due to zero-inflation in the heatwave treatments, corrections were applied by fitting a different error distribution (producing theta ~1)[66]. For moderate over-dispersion (1 < theta < 20), a quasi-Poisson error with a log link function was fitted. For strong over-dispersion (theta > 20), a negative binomial model with log link was fitted using 'glm.nb{MASS}'[89] (see Table 1 and Supplementary Table 1 for model errors and link functions). Continuous variables (mating duration and GFP sperm density distributions) were initially fitted using a Gaussian distribution with an identity link function, however, both had positively skewed residuals and outliers[66]. Model fits were improved for mating duration by using a log link function. Proportion response variables, which included paternity share in sperm competitions, sperm viability, and hatching, pupation and eclosion success, were fitted using a binomial distribution and a logit link function. Response variables were entered as a two column matrix of success-and-fail using cbind(success, fail){base}[66]. Where over-dispersion was present, usually due to zero-inflation in the heatwave treatments, it was corrected for by fitting a quasi-binomial distribution with a logit link function[66]. (See Table 1 and Supplementary Table 1 for model errors and link functions).

After each maximal model was fitted, the statistical significance of the experimental treatment variables were assessed using Akaike's Information Criterion (AIC) comparisons, and log likelihood ratio tests (LLRT) with, and without, the term of interest[83]. The most efficient models had significantly lower AICs[90,91]. LLRTs were $\chi^2$ tests when the response variable was a count or proportion, and F tests when continuous[66]. LLRTs were primarily computed with 'drop1{stats}'[66,78,83]. 'drop1{stats}' was not compatible with quasi-error distributions, so was substituted for 'lrtest{lmtest}'[92]. Simple post-hoc comparisons between treatment groups and controls were derived from summary(model)[85,93]. Post-hoc pairwise Tukey comparisons were applied using 'lsmeans{lsmeans}'[94]. As a measure of how much variation in the response variable was explained by the model, pseudo $R^2$ (explained deviance) was calculated for GLMs[66]. For GLMMs, 'r. squaredGLMM{MuMIn}'[95] reported the marginal $R^2$ explained by the fixed factors, and conditional $R^2$ for the fixed and random factors.

## Data availability
All source data generate and analysed in this study, and which underlie all results Figures in the Main and Supplementary Information sections of the Article, are provided as an associated Source Data file, or are available directly from the authors. Raw data and R codes are available through Dryad at https://doi.org/10.5061/dryad.846st51. A reporting summary for this Article is also available as a Supplementary Information file.

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

## Acknowledgements

This work was supported by NERC (Grant NE/K013041/1 and the EnvEast DTP), the Leverhulme Trust and the University of East Anglia. We thank Scott Pitnick and John Belote for GFP lines, and Jessie Gardner, Lewis Spurgin, Will Nash and Damian Smith for advice and help which improved the study.

## Author contributions

M.J.G.G., K.S. and M.E.D. conceived and designed the study, with input from all authors, including A.F. K.S., M.E.D. and R.V. led the experimental assays, with input from AL in the sperm competition experiment, M.J.G.G., P.T. and Ł.M. in the sperm analyses, J.L.G. and L.H. in the transgenerational assays, and all authors contributed to culture and

maintenance. K.S. led the data analyses, with input from J.L.G. M.J.G.G. and K.S. wrote the paper, with contributions from all authors.

**Additional information**

**Competing interests:** The authors declare no competing interests.

