## [Peer Review File · Nature Communications]

Reviewers' Comments:

Reviewer #1:

Remarks to the Author:

In the ms 'Heatwaves compromise sperm function and cause transgenerational damage in a model insect' submitted by Sales et al., the authors report on their investigation on the effects a heatwave has on male reproduction. They investigate several determinants of male reproductive success and found them all to be negatively affected by heat and to severely damage male reproductive capabilities and even extend to damage offspring. This is an important and timely question and these data are lacking.

The ms is well written, the experiments described carefully and in great detail and the analyses performed and described with equal care.

I have a few points I would like to remark and ask the authors to consider:

The transgenerational impact is fascinating and maybe even unexpected. The authors speculate that epigenetic changes could be one explanation for their findings. Are there any reports that sperm epigenetics can still be altered in mature sperm stored in the female sperm storage organs? Also Agrawal and Wang 2008 PLoS Biology reported that in *Drosophila* females are able to repair DNA damage of sperm while it resides in the female reproductive tract? Here the opposite seems to be true and sperm show higher damage due to a heatwave than when males are exposed before mating. Any ideas what might explain the different sensitivities of sperm depending on their environment?

Regarding the analyses on line 727 and following you describe to only use an F-distribution for continuous response variables. According to Crawley 2007 (e.g. page 522) when the quasi-extension is used then significance should also be estimated by comparison to an F-distribution and not against a Chi-square distribution.

The authors might want to consider Rohmer et al. 2004 Journal of Experimental Biology, where they studied the effect of temperature on spermatogenesis and sperm abnormalities in *Drosophila melanogaster*.

Regarding heat-shocking the GFP labeled sperm, can the heat shock affect the intensity and longevity of the GFP and could that disrupt your signal to correctly describe the distribution of sperm inside the female reproductive tract?

When using the sperm viability kit, how did you check that a substantial fraction of the sperm were stained and the procedure worked and that the same fraction in all samples were stained? Was there a control for the efficiency of the staining reaction?

How well did your incubators regulate temperature, what was the variability in those? Did you measure temperature fluctuations during your heatwaves?

Line 420: please rewrite this sentence

Supplementary Table S1 needs to split over several pages/tables and the size increased. In the current stage it is unreadable and also using a larger magnification does not make it nicer and makes more readable.

Reviewer #2:

Remarks to the Author:

Climate change is continuing apace, and as this past year has shown, temperatures are hitting highs that are increasingly extreme. The full extent of these deviations are unclear, but we know there are major risks to biodiversity and accelerating extinction risks. This includes problems in the oceans for example where male fertility seems especially sensitive to climate variation.

This submission focuses on temperature effects on male fertility and sperm function in a beetle (on land) and I note that beetles to a first approximation are all animals (Haldane had a nice quote about this), thus the work is general and important. It is also extremely timely given the background above and represents a very comprehensive set of experiments that dissects effects to a depth not previously seen in this sort of investigation, including using GFP labeled sperm.

The results seem pretty clear cut and are sure to be of very broad interest, from reproductive physiologists to evolutionary and behavioral biologists to conservationists and beyond. I have some questions around the analyses, which are not clear to me and a more general question that the authors may wish to consider.

To the analysis. For the sex by temperature model (Figure 1), it seems this is a pretty straight forward GLM with 2 levels in sex and 6 levels in temperature and you report a sex by temperature interaction (Table 1). Figure 1 does not produce an interaction plot and shows significances of within temperature between sex effects (figure legend). I do not quite understand what has been done here? If there is a sex by temperature effect then why not just plot it and describe it? Sorry but I am confused about exactly what you are saying here. Why does the overall significance indicated on the figure not relate to the interaction?

In figure 2, the red box was not clear in my download and you have two female heat-wave treatments (Figure 2a) so to call one a control is not quite right in the context of what control means elsewhere. I can see what you mean, but it is not a control like the rest of the figure and this is kind of confusing. And at 2c you again show an overall p (which I guess is the model p: table 1) and then the post-hoc test with letters. Given that the table has the overall model effect why not just refer to it and show the post-hoc differences here – again just to make it easy for the reader. And ditto with other figures too. In Table 1 you sometimes have 2 or sometimes 3 paternal heatwave levels (1 v 2 DF) but it is not clear why, and then on top of that sometimes 1, 5, 6 DF for male heat-wave. Again this is not clear why. It would be good to explain all this to help the reader understand what you have done exactly. And in the methods you refer to 6 heat treatments then link to Figure 4c which shows a comparison between only 2 of them (Line 440). Again this needs checking/clarifying.

Finally for the survival analysis, you say randomly sexed offspring were used: did you check to ensure that either treatment did not have more of one sex or the other in it as this could bias outcomes (e.g. if heatwave contained mostly males).

My general question is do you think sperm are especially susceptible to stress or is it temperature that is the danger? I wonder if you would care to comment in the manuscript?

Minor points

Line 690: what is a sampling structure variable – do you mean block?

Line 3: can a species decline? Is it clearer to say population sizes are declining?

Reviewer #3:

Remarks to the Author:

This manuscript examines the effect of simulated heatwave conditions (5 days at 5 °C above optimum temperature) on reproductive traits of the red flour beetle *Tribolium castaneum*. The work is extremely thorough, comprising a series of experiments that not only explore the consequences of heatwaves for reproduction in males and females, but also identify the underlying mechanisms, and transgenerational effects. The authors demonstrate deleterious effects of heat stress for mating

success and reproductive output in males (but not females), and similar effects on sperm stored in female sperm storage organs. They also found transgenerational effects of heatwave conditions: there was reduced reproductive potential of male offspring, and longevity of all offspring sired by males or sperm exposed to heatwave conditions, relative to those sired by controls. They demonstrate that the decline in reproductive output of males exposed to heatwave conditions can be explained by a decline in the quantity, viability, and transfer and storage of sperm.

The paper is well-written, and I enjoyed reading it. The methods and supplementary materials are very comprehensive, and provide all of the necessary detail to allow the experiments to be reproduced. The statistical analyses seem robust and appropriate, and there is very clear reporting of sample sizes and data distributions.

Effects of thermal stress on organism fitness have received a lot of attention in recent years, due to interest in predicting the consequences of climate change and explaining species' declines. There is a sizeable body of work demonstrating detrimental effects of heat stress on male reproductive success. One minor criticism I had was that the authors imply that studies demonstrating these effects in ectotherms have been confined to *Drosophila melanogaster*. While I agree that ectotherms have been neglected relative to mammals and other endotherms, previous studies have covered a broader array of taxa than is suggested, and it would be nice to see this acknowledged. As examples, I have identified a few additional references below.

Nevertheless, this manuscript presents several exciting, novel findings that go beyond previous studies. Specifically, the observations that (i) exposure of mated females (i.e. carrying stored sperm) to heatwave conditions reduced reproductive success in a similar way to exposure of males, even though unmated females showed no detrimental effects of heat stress on reproduction; and (ii) exposure of males or sperm stored in female storage organs to heatwave conditions had a detrimental effect on longevity and reproduction in offspring. I know of one previous study that has reported transgenerational effects of male heat stress on offspring longevity in an insect (Gasparini et al. 2017; see below), but to my knowledge none have demonstrated transgenerational effects via exposure of sperm stored in the female storage organs.

These two results in particular have major implications for predicting the consequences of rising temperatures on population viability. They are likely to be of interest to both specialists in this field and more broadly across the biological sciences, and to influence the direction of future work on the topic.

Suggested additional references:

Deleterious effects of heat stress on male reproductive success in ectotherms:

Breckels & Neff (2013). The effects of elevated temperature on the sexual traits, immunology and survivorship of a tropical ectotherm. *The Journal of Experimental Biology* 216, 2658 – 2664

Zeh et al. (2012). Degrees of disruption: projected temperature increase has catastrophic consequences for reproduction in a tropical ectotherm. *Global Change Biology* 18, 1833 – 1842

Zizzari & Ellers (2011). Effects of exposure to short-term heat stress on male reproductive fitness in a soil arthropod. *Journal of Insect Physiology* 57, 421 - 426

Transgenerational effects of heat stress on male reproductive traits:

Gasparini et al. (2017). Paternal-effects in a terrestrial ectotherm are temperature dependent but no evidence for adaptive effects. *Functional Ecology* 32, 1011 – 1021

I have provided a few specific comments, questions and suggestions on the manuscript below:

- Page 5, L108: 'acclimation' would be a more accurate word than 'adaptation' to use here.
- In the experiment testing the effect of reproductive output of males exposed to either one or two heatwaves (vs controls), males were age-matched at the time of measuring reproductive output. However, males exposed to two heatwaves experienced their first heatwave at a younger age than those experiencing one heatwave. Does the sensitivity of males to heatwaves change with age? i.e. could the decline in reproductive output of males exposed to two heatwaves be at least partially explained by having had their first exposure at a more sensitive age?
- Sperm traits were not reported for mated females exposed to heatwave conditions, which also showed a decline in reproductive success that you attribute to effects on stored sperm. Do you know whether sperm exposed to heatwaves within the female storage organs also has reduced viability, or could this observation be explained by different processes than the effects on males e.g. increased sensitivity of mated females to thermal stress?
- Is there potential for female choice to compensate for negative effects of heat shock, by choosing to mate with males who have not been exposed to heat stress, or suffer less cost of exposure?
- How much opportunity is there for behavioural thermoregulation in *Tribolium* in their natural environment? Can individuals seek out less stressful microclimates, which may limit these deleterious effects? It would be particularly interesting to know whether mated and unmated females differ in their selection of thermal environment, given the very different consequences of heatwave exposure.
- Temperature variability is also expected to rise with climate change and at least two recent studies have shown that temperature variability has detrimental effects on male reproduction (references provided below). In these cases, there is evidence that this effect is because exceeding critical thermal thresholds even for short periods each day is sufficient to reduce male reproductive success. Therefore, although the heatwave conditions you use seem quite extreme, the effects could be evident even with much shorter periods of exposure. This may be worth mentioning to further emphasise the potential consequences for population viability.

References showing deleterious effects of temperature variability on male reproductive success:

Zeh et al. (2014). Constant diurnal temperature regime alters the impact of simulated climate warming on a tropical pseudoscorpion. *Scientific Reports* 4, 3706

Saxon et al. (2018). Temperature fluctuations during development reduce male fitness and may limit adaptive potential in tropical rainforest *Drosophila*. *Journal of Evolutionary Biology* 31, 405 - 415

Heatwaves compromise sperm function and cause transgenerational damage in a model insect - NCOMMS-18-24197-T

Response to Reviewers' comments:

Reviewer #1 (Remarks to the Author):

In the ms ,Heatwaves compromise sperm function and cause transgenerational damage in a model insect' submitted by Sales et al., the authors report on their investigation on the effects a heatwave has on male reproduction. They investigate several determinants of male reproductive success and found them all to be negatively affected by heat and to severely damage male reproductive capabilities and even extend to damage offspring. This is an important and timely question and these data are lacking. The ms is well written, the experiments described carefully and in great detail and the analyses performed and described with equal care.

RESPONSE: Sincere thanks for the positive overall assessment, and for the detailed, thoughtful review.

I have a few points I would like to remark and ask the authors to consider: The transgenerational impact is fascinating and maybe even unexpected. The authors speculate that epigenetic changes could be one explanation for their findings. Are there any reports that sperm epigenetics can still be altered in mature sperm stored in the female sperm storage organs? Also Agrawal and Wang 2008 PLoS Biology reported that in Drosophila females are able to repair DNA damage of sperm while it resides in the female reproductive tract? Here the opposite seems to be true and sperm show higher damage due to a heatwave then when males are exposed before mating. Any ideas what might explain the different sensitivities of sperm depending on their environment?

RESPONSE: We agree that the transgenerational results were surprising (and also worrying). Because we ran our transgenerational experiments exploring heatwave impacts on males versus sperm-in-female storage separately, we do not feel confident that we can generally conclude a stronger transgenerational impact of thermal stress on sperm-in-female-storage versus the impact in males. Comparing figures 4 c versus e, and 4d versus f, it is not clear that a much stronger transgenerational impact occurs for heatwaves on sperm-in-storage compared with treatments of paternal males, and there is some variation in control data between experiments. However, we agree it relevant to include the information that oocytes and developing zygotes can repair sperm DNA after fertilization, so now include the following statement:

Line 247: Oocyte and zygote cell repair mechanisms can reverse sperm DNA damage through embryogenesis⁶¹, but this may not be possible following our heatwave treatment conditions if the DNA damage is sufficiently severe.

Regarding the analyses on line 727 and following you describe to only use an F-distribution for continuous response variables. According to Crawley 2007 (e.g. page 522) when the quasi-extension is used then significance should also be estimated by comparison to an F-distribution and not against a Chi-square distribution.

RESPONSE: According to Thomas et al. 2015 (Data analysis with R Statistical Software. Cardiff: Eco-explore), F distributions are recommended for continuous variables and Chi-square distributions for categorical or integer dependent variables such as proportions and counts (e.g. page 62, Thomas et al. 2015). We have therefore followed the Thomas et al (2015) recommendations, including the explicit use of Chi-square for quasi-binomial distributions (bottom line of page 96, Thomas et al. 2015), but accept that different experts can recommend differing approaches. We would prefer to retain our current analytical error distribution approach, but could re-analyse according to F-distributions too. However, please also note that 1) effects and differences are clear so no conclusions would be changed, and 2) our manuscript already contains very large and detailed levels of statistical information.

The authors might want to consider Rohmer et al. 2004 Journal of Experimental Biology, where they studied the effect of temperature on spermatogenesis and sperm abnormalities in *Drosophila melanogaster*.

RESPONSE: We agree this is a highly relevant publication, and it was cited in the previous version of the manuscript (reference number 27).

Regarding heat-shocking the GFP labeled sperm, can the heat shock affect the intensity and longevity of the GFP and could that disrupt your signal to correctly describe the distribution of sperm inside the female reproductive tract?

*RESPONSE: We can find no evidence that GFP excitation will decrease following sperm heat stress, and it was obvious throughout the imaging assays that observations of sperm density (under phase contrast microscopy) were closely associated with GFP intensity. Previous studies have reported GFP fluorescence to be stable at temperatures above 60°C (e.g. Tsien, R.Y. The Green Fluorescent Protein. Ann. Rev. Biochem. **67**, 509–544 (1998); Ishii, M. et al Study on the thermal stability of green fluorescent protein (GFP) in glucose parenteral formulations. Int. J. Pharmac. **337**, 109–117 (2007)). Our samples were also dissected and imaged within two weeks of flash freezing and storage at -80°C, whereas pilot tests revealed GFP still being clearly visible after several months of storage in dead sperm within our system. Despite all this, and as suggested by Reviewer 1, we cannot be sure that reduced GFP intensity is not also associated with sperm cell damage / death in the female tract, so add the following statement (in bold):*

Line 166: The amount of GFP-labelled sperm present in these sites 24h after mating was reduced by two-thirds when females had mated with males previously exposed to 42°C heatwave conditions (Figure 3b, d, e), either as a consequence of lower sperm densities,

or due to reduced GFP excitation associated with dying sperm.

When using the sperm viability kit, how did you check that a substantial fraction of the sperm were stained and the procedure worked and that the same fraction in all samples were stained? Was there a control for the efficiency of the staining reaction?

RESPONSE: *We used three different channels for observing sperm in the viability assay: 1) red and 2) green fluorescence channels, and 3) DIC (for detecting non-stained sperm). All sperm observed in the viability assay stained either red or green under, so the reaction was complete in this experiment and no additional control was required. We now add this detail to the methods:*

Line 628: Sperm heads were imaged in 1) red and 2) green fluorescence channels, and 3) Differential Interference Contrast (for detecting non-stained sperm). All sperm observed in the viability assay took up the stain to fluoresce either red or green.

How well did your incubators regulate temperature, what was the variability in those? Did you measure temperature fluctuations during your heatwaves?

RESPONSE: *We used avian egg incubators (Octagon 20s, Brinsea Ltd, Line 369) which maintain very tight thermal regulation, with heatwave conditions exposed to beetle groups held in petri dishes across the central plane of the incubator. Thermal variation did not exceed 1°C above or below the set point throughout the heatwave treatment. We now add further information to the methods:*

Line 375: Beetles were exposed to heatwaves in single-sex groups of 20 individuals in 5cm Petri dishes containing standard fodder and positioned in the central plane of the incubator. Temperatures did not exceed 1°C above or below the treatment set point, checked using a 35°C to 45°C mercury incubation thermometer (G.H. Zeal Ltd, Zeal House, 8 Deer Park Road, London, SW19 3UU, U.K.) calibrated to United Kingdom Accredited Service standards (Charnwood Instrumentation Services Ltd, 81 Park Road, Coalville, Leicestershire, LE67 3AF, U.K.).

Line 420: please rewrite this sentence

RESPONSE: *We have broken up and re-written the sentence:*

Line 463: To measure the impact of additional heatwaves, adult males were exposed to three treatments: 1) Control: five days of exposure to 30°C (n=20); 2) Single heatwave: five days of heatwave exposure at 42°C (n=35); and 3) Double heatwaves: five days of heatwave exposure at 42°C followed by ten days at 30°C followed by a second five days of heatwave exposure at 42°C (n=29).

Supplementary Table S1 needs to split over several pages/tables and the size increased.

In the current stage it is unreadable and also using a larger magnification does not make it nicer and makes more readable.

***RESPONSE:** Agreed, we have lengthened and reformatted it so it now crosses two pages and is more readable.*

Reviewer #2 (Remarks to the Author):

Climate change is continuing apace, and as this past year has shown, temperatures are hitting highs that are increasingly extreme. The full extent of these deviations are unclear, but we know there are major risks to biodiversity and accelerating extinction risks. This includes problems in the oceans for example where male fertility seems especially sensitive to climate variation.

This submission focuses on temperature effects on male fertility and sperm function in a beetle (on land) and I note that beetles to a first approximation are all animals (Haldane had a nice quote about this), thus the work is general and important. It is also extremely timely given the background above and represents a very comprehensive set of experiments that dissects effects to a depth not previously seen in this sort of investigation, including using GFP labeled sperm.

The results seem pretty clear cut and are sure to be of very broad interest, from reproductive physiologists to evolutionary and behavioral biologists to conservationists and beyond. I have some questions around the analyses, which are not clear to me and a more general question that the authors may wish to consider.

RESPONSE: Sincere thanks for the positive overall assessment, and for the detailed, thoughtful review.

To the analysis. For the sex by temperature model (Figure 1), it seems this is a pretty straight forward GLM with 2 levels in sex and 6 levels in temperature and you report a sex by temperature interaction (Table 1). Figure 1 does not produce an interaction plot and shows significances of within temperature between sex effects (figure legend). I do not quite understand what has been done here? If there is a sex by temperature effect then why not just plot it and describe it? Sorry but I am confused about exactly what you are saying here. Why does the overall significance indicated on the figure not relate to the interaction?

RESPONSE: Indeed this first result is a GLM with 2 sexes and 6 thermal treatments, analysis details are reported in Results, Table 1, and with further information in Table S1. We believe that plotting the data using boxplots, where male and female responses are side-by-side, is the best compromise (supported by the analysis detail) for presenting as much graphical information from this experiment in the most simple of terms. Although the interaction is important, so are the responses (or not) within either sex (hence the significant differences defined by letters on the figure). We considered an interaction plot, but concluded it would be less informative. First, because the analysis defines the thermal treatments as categorical variables, it would not be appropriate to plot the means (or medians) through a continuous range (unless we then extended the X axis to reflect the temperature differences between the six treatments). There is statistical debate about whether ~six groups should be defined as categorical or continuous variables (e.g. McDonald, J.H. Handbook of Biological Statistics 2nd ed., Maryland: Sparky House Publishing (2009)). Second, an interaction plot would lose a lot of very

useful information that is currently detailed on the boxplot concerning the spread of data within each treatment, as well as variation across means, medians and IQRs. This information is particularly important because many readers will wish to know the distribution of zeros within a treatment, which informs on biological responses to heatwaves. We also considered presenting the interaction using the boxplot style, but where male and female results are paired within each of the six thermal treatments, but this produces a more confusing and very dense graph. We concluded that the two boxplots side-by-side is a good compromise between showing graphically the interaction (statistically detailed in the results and Tables), while presenting fully the spread of data within each treatment group and sex.

In figure 2, the red box was not clear in my download and you have two female heat-wave treatments (Figure 2a) so to call one a control is not quite right in the context of what control means elsewhere. I can see what you mean, but it is not a control like the rest of the figure and this is kind of confusing.

***RESPONSE:** Apologies that the colours were not clear in the download and we hope they now are (with heatwave treatments identified using orange boxes throughout, and controls white). We agree that Figure 2a created confusion by having both boxes coloured orange, even though one was labeled 'control', so have altered this in line with suggestions. Figure 2a is now labeled as 'Sperm treatment in female', to focus on the treatment experienced by sperm within the female: either control (no heat) or heatwave (5 days at 42°C). To assure readers that our 'control' was a female that had also experienced a heatwave, but before insemination (and therefore no heatwave exposure to sperm), we slightly elaborate the legend:*

*Line 302: **α**, Impacts of heatwaves on inseminated sperm: reproductive output of females exposed to heatwaves before mating and sperm storage (Control: n=55) compared to females exposed to heatwaves after mating and with inseminated sperm in storage (Heatwave: n=62).*

And at 2c you again show an overall p (which I guess is the model p: table 1) and then the post-hoc test with letters. Given that the table has the overall model effect why not just refer to it and show the post-hoc differences here – again just to make it easy for the reader. And ditto with other figures too.

***RESPONSE:** We would prefer to retain the presentation of overall model p values on the figures as asterixes, and to present post-hoc differences (where relevant and there are 3+ groups) using standard lettering. We believe this maximizes the information to the reader using the very simplest notations. More statistical detail including model summaries can then be reviewed in Table 1, with even more detail in Supplementary Table S1. We believe this approach allows access to all the information across a hierarchy of detail, without swamping figures with analytical detail.*

In Table 1 you sometimes have 2 or sometimes 3 paternal heatwave levels (1 v 2 DF) but it is not clear why, and then on top of that sometimes 1, 5, 6 DF for male heat-wave. Again this is not clear why. It would be good to explain all this to help the reader understand what you have done exactly.

RESPONSE: *Table 1 indeed presents the number of treatments analysed within each experiment, according to the number of degrees of freedom. Cross-referencing to the figures explains what these 2, 3 or 6 treatments are, and much more information is accessible in Supplementary Table 1 (including exact treatment detail). To improve explanation, we now add the relevant Figure number to the 'Experiment' column on Table 1, and we have expanded Table S1 to split between two pages (landscape format) make it more readable.*

And in the methods you refer to 6 heat treatments then link to Figure 4c which shows a comparison between only 2 of them (Line 440). Again this needs checking/clarifying.

RESPONSE: *Apologies, this was our error. The reference should be to (SUPPLEMENTARY) Figure S4c, not Figure 4c, which we have now corrected in the Methods. We have also amended Supplementary Figure 4 (a) to indicate that the heatwave condition treatments could be UP TO 42°C.*

Finally for the survival analysis, you say randomly sexed offspring were used: did you check to ensure that either treatment did not have more of one sex or the other in it as this could bias outcomes (e.g. if heatwave contained mostly males).

RESPONSE: *A good point; we have checked offspring sex ratios following male heatwave conditions at 42°C and found no evidence for sex bias. Of 17 crosses, the average % of males among offspring batches was 51%. We now include the following statement in the revised manuscript:*

Line 654: Previous measures showed that sex ratios within offspring groups sired by males previous exposed to heatwave conditions did not depart from unity: average % male across n=17 offspring groups = 51% (± 2.36); Wilcoxon test of male proportion versus 0.5: $V_{17}=79$; $P=0.59$.

My general question is do you think sperm are especially susceptible to stress or is it temperature that is the danger? I wonder if you would care to comment in the manuscript?

RESPONSE: *We agree that there is evidence that sperm function is generally sensitive to abiotic and/or biotic stress, so broaden interpretation in the revised manuscript, adding the following statement and three relevant references:*

Line 194: Although our experiments focus on temperature, spermatozoa are among the

most complex and diverse eukaryotic cell types⁴⁸, with functional sensitivities to physiological⁴⁹ and genetic⁵⁰ stress, so our findings may also be due to a general spermatozoal susceptibility to stress. In addition, we exposed adult beetles to temperature increases for five days in order to replicate heatwave conditions, however, recent work has shown that thermal impacts on reproduction can also occur over relatively short windows of more acute exposure^{51,52}. Whichever of these situations apply, our combined findings could shed light on why populations have declined as a result of increased thermal or general stress from climate change^{4-7,12,22,23}.

Line 690: what is a sampling structure variable – do you mean block?

RESPONSE: *Yes, we mean a block (or experimental repeat), and now clarify:*

Line 759: Where sampling structure variables (= blocks or experimental repeats) were present

Line 3: can a species decline? Is it clearer to say population sizes are declining?

RESPONSE: *We now change ‘species’ to ‘populations’.*

Reviewer #3 (Remarks to the Author):

This manuscript examines the effect of simulated heatwave conditions (5 days at 5 °C above optimum temperature) on reproductive traits of the red flour beetle *Tribolium castaneum*. The work is extremely thorough, comprising a series of experiments that not only explore the consequences of heatwaves for reproduction in males and females, but also identify the underlying mechanisms, and transgenerational effects. The authors demonstrate deleterious effects of heat stress for mating success and reproductive output in males (but not females), and similar effects on sperm stored in female sperm storage organs. They also found transgenerational effects of heatwave conditions: there was reduced reproductive potential of male offspring, and longevity of all offspring sired by males or sperm exposed to heatwave conditions, relative to those sired by controls. They demonstrate that the decline in reproductive output of males exposed to heatwave conditions can be explained by a decline in the quantity, viability, and transfer and storage of sperm.

The paper is well-written, and I enjoyed reading it. The methods and supplementary materials are very comprehensive, and provide all of the necessary detail to allow the experiments to be reproduced. The statistical analyses seem robust and appropriate, and there is very clear reporting of sample sizes and data distributions.

RESPONSE: Sincere thanks for the positive overall assessment, and for the detailed, thoughtful review.

Effects of thermal stress on organism fitness have received a lot of attention in recent years, due to interest in predicting the consequences of climate change and explaining species' declines. There is a sizeable body of work demonstrating detrimental effects of heat stress on male reproductive success. One minor criticism I had was that the authors imply that studies demonstrating these effects in ectotherms have been confined to *Drosophila melanogaster*. While I agree that ectotherms have been neglected relative to mammals and other endotherms, previous studies have covered a broader array of taxa than is suggested, and it would be nice to see this acknowledged. As examples, I have identified a few additional references below.

RESPONSE: We agree we did not include all reference to work on thermal stress and reproduction in ectotherms identified by Reviewer 3, which was largely due to concerns about word count and limited space in the Nature Comms format. As advised by Reviewer 3, we now include four further references of relevance to the revised manuscript (References 39 to 42):

- Zizzari & Ellers (2011). Effects of exposure to short-term heat stress on male reproductive fitness in a soil arthropod. Journal of Insect Physiology 57, 421 – 426.*
- Zeh et al. (2012). Degrees of disruption: projected temperature increase has catastrophic consequences for reproduction in a tropical ectotherm. Global Change*

Biology 18, 1833 – 1842.

- Breckels & Neff (2013). The effects of elevated temperature on the sexual traits, immunology and survivorship of a tropical ectotherm. The Journal of Experimental Biology 216, 2658 – 2664.

- Gasparini et al. (2017). Paternal-effects in a terrestrial ectotherm are temperature dependent but no evidence for adaptive effects. Functional Ecology 32, 1011 – 1021.

Nevertheless, this manuscript presents several exciting, novel findings that go beyond previous studies. Specifically, the observations that (i) exposure of mated females (i.e. carrying stored sperm) to heatwave conditions reduced reproductive success in a similar way to exposure of males, even though unmated females showed no detrimental effects of heat stress on reproduction; and (ii) exposure of males or sperm stored in female storage organs to heatwave conditions had a detrimental effect on longevity and reproduction in offspring. I know of one previous study that has reported transgenerational effects of male heat stress on offspring longevity in an insect (Gasparini et al. 2017; see below), but to my knowledge none have demonstrated transgenerational effects via exposure of sperm stored in the female storage organs.

These two results in particular have major implications for predicting the consequences of rising temperatures on population viability. They are likely to be of interest to both specialists in this field and more broadly across the biological sciences, and to influence the direction of future work on the topic.

***RESPONSE:** Sincere thanks for the positive views, and we now rightly add reference to Gasparini et al (2017) in the manuscript, together with the following discussion:*

*Line 233: In a recent study exposing field crickets (*Gryllus bimaculatus*) to 24°C and 28°C regimes, the warmer treatment when exposed to adult males was found to reduce ejaculate sperm number (with the reverse effect seen when 28°C exposure took place through the pre-adult stages as well)⁴². Adult males exposed to the warmer 28°C regime also fathered offspring that exhibited reduced survival (and again the reverse effect was seen with improved offspring survival if warmer 28°C exposure occurred throughout development)⁴².*

I have provided a few specific comments, questions and suggestions on the manuscript below:

- Page 5, L108: ‘acclimation’ would be a more accurate word than ‘adaptation’ to use here.

***RESPONSE:** ‘Adaptation’ replaced with ‘acclimation’ as suggested (now at Line 109).*

- In the experiment testing the effect of reproductive output of males exposed to either one or two heatwaves (vs controls), males were age-matched at the time of measuring

reproductive output. However, males exposed to two heatwaves experienced their first heatwave at a younger age than those experiencing one heatwave. Does the sensitivity of males to heatwaves change with age? i.e. could the decline in reproductive output of males exposed to two heatwaves be at least partially explained by having had their first exposure at a more sensitive age?

RESPONSE: Apologies for an incorrect description of our double heatwave experiment in the text of our previous manuscript, but thanks for identifying. As we described (correctly) in the previous methods Figure S7, all males were the same age (12 ± 2 days post eclosion) when they received their first experimental treatment, with males in the double heatwave group then receiving their second heatwave 10 days later. Although this meant that males in the double heatwave group were introduced to females at 27 ± 2 days post eclosion, compared with 17 ± 2 days post eclosion for the single heatwave and control groups, the timescale and protocol ensured that all males were reproductively mature (within 10 days post eclosion), thereby limiting any differential effects of development and/or early spermatogenesis. We change the text accordingly:

Line 483: To minimise developmental effects through initial spermatogenesis, all males were reproductively mature (12 ± 2 days post eclosion) and received their initial 5-day treatments simultaneously, with males in group 3) experiencing their second heatwave at age 27 ± 2 days post eclosion. Thus, all males were reproductively mature when exposed to single or double heatwaves (Figure S7).

- Sperm traits were not reported for mated females exposed to heatwave conditions, which also showed a decline in reproductive success that you attribute to effects on stored sperm. Do you know whether sperm exposed to heatwaves within the female storage organs also has reduced viability, or could this observation be explained by different processes than the effects on males e.g. increased sensitivity of mated females to thermal stress?

RESPONSE: Unfortunately we were not able to measure sperm viability once stored within females because of the technical challenges of accessing/recovering/staining sperm once held within female tract storage. Tribolium castaneum adults are only ~3mm long, so many sperm assays are a technical challenge, but sperm within storage are extremely difficult to recover or stain in a consistent and controlled manner.

- Is there potential for female choice to compensate for negative effects of heat shock, by choosing to mate with males who have not been exposed to heat stress, or suffer less cost of exposure?

RESPONSE: In all our assays females (and males) were not given the opportunity to choose between mates, but allocated an experimental partner according to their treatment group. Tribolium castaneum is a relatively promiscuous species, and we found no evidence that females chose not to mate with heat-stressed males: of 36 females

paired with previously heat-stressed males, all had successfully mated within one hour of pairing (Line 127). There is obvious potential for females to use mate choice and/or strategic polyandry to guard against male-derived infertility as a consequence of heatwave damage, and we are currently exploring this possibility. We now include a consideration of female choice to our suggestions for future research in this area in the revised manuscript:

Line 252: We urge future research into the molecular basis of this transgenerational heatwave damage, whether females have evolved mate choice strategies to avoid male-derived thermo-sensitive infertility, and for the consequences of our findings of heatwave damage to male reproductive function to be examined in a broader range of taxa.

- How much opportunity is there for behavioural thermoregulation in *Tribolium* in their natural environment? Can individuals seek out less stressful microclimates, which may limit these deleterious effects? It would be particularly interesting to know whether mated and unmated females differ in their selection of thermal environment, given the very different consequences of heatwave exposure.

***RESPONSE:** Escaping thermal stress by seeking microclimatic variation will be important for any mobile animal in the natural environment, including *Tribolium castaneum* when it occupies grain stores or travels between them. We have not studied this behaviour in *T. castaneum* because we wanted to understand the impact of thermal stress on reproduction when even microclimates present inescapable 40 to 42°C heatwave conditions. As we report in the manuscript (Line 368) ‘These conditions have been recorded in the natural environment across more than 90 countries⁶⁸.’*

- Temperature variability is also expected to rise with climate change and at least two recent studies have shown that temperature variability has detrimental effects on male reproduction (references provided below). In these cases, there is evidence that this effect is because exceeding critical thermal thresholds even for short periods each day is sufficient to reduce male reproductive success. Therefore, although the heatwave conditions you use seem quite extreme, the effects could be evident even with much shorter periods of exposure. This may be worth mentioning to further emphasise the potential consequences for population viability.

***RESPONSE:** We agree that shorter-term thermal exposure could also cause the effects we see, but wanted to retain our standard treatment to mimic that of a defined ‘heatwave’ (= >5days at >5°C above average). We now add this statement to our interpretations in the revised manuscript:*

Line 198: In addition, we exposed adult beetles to temperature increases for five days in order to replicate heatwave conditions, however, recent work has shown that thermal impacts on reproduction can also occur over relatively short windows of more acute

exposure^{51,52}. Whichever of these situations apply, our combined findings could shed light on why populations have declined as a result of increased thermal or general stress from climate change^{4-7,12,22,23}.

References showing deleterious effects of temperature variability on male reproductive success:

Zeh et al. (2014). Constant diurnal temperature regime alters the impact of simulated climate warming on a tropical pseudoscorpion. *Scientific Reports* 4, 3706

Saxon et al. (2018). Temperature fluctuations during development reduce male fitness and may limit adaptive potential in tropical rainforest *Drosophila*. *Journal of Evolutionary Biology* 31, 405 - 415

RESPONSE: *Thanks, and these are included as references 51 and 52 (at Line 201).*

Reviewers' Comments:

Reviewer #1:

Remarks to the Author:

I thank the authors for their considerations of all my comments given and integrating them in a revised version of the ms as appropriate. I fully recommend the ms for publication.

Reviewer #2:

Remarks to the Author:

Thanks for the very comprehensive responses and for clarifying issues that were not clear to me in the initial submission. My major concern was addressed - the relevant figure was in the Supplementary information rather than the main text. I think the revised submission is now suitable for publication and am confident the work will be very well received.

I have no further questions. Congratulation on an excellent and interesting study.

Reviewer #3:

Remarks to the Author:

The authors have thoroughly addressed the comments of all of the reviewers in their revision. I am satisfied with the revised version of the manuscript and have no further suggestions for change.